# Peritoneal Dialysis as a Renal Replacement Therapy Modality for Patients with Acute Kidney Injury

**DOI:** 10.3390/jcm11123270

**Published:** 2022-06-08

**Authors:** Sana Farooq Khan

**Affiliations:** Division of Nephrology, University of Virginia Health System, Charlottesville, VA 22908, USA; sk4yp@virginia.edu; Tel.: +1-434-924-5125

**Keywords:** acute kidney injury, peritoneal dialysis

## Abstract

Since the advent and predominant use of extracorporeal therapies for renal replacement therapies for acute kidney injury, the use of peritoneal dialysis has largely been limited to specific resource-limited settings. This review highlights the current data available for the utilization of peritoneal dialysis for acute kidney injury. Though the current randomized controlled trials have small patient numbers, they have demonstrated peritoneal dialysis to be an appropriate modality for dialysis therapy in acute kidney injury. Current outcomes do not show a difference in mortality, renal recovery rates, or infectious complications when compared to extracorporeal treatments. However, there is a marked heterogeneity in these trials, and more standardized reporting of trial design, techniques, complications, and outcomes is needed.

## 1. Introduction

Acute kidney injury (AKI) is increasingly prevalent globally and is associated with high morbidity, mortality, and cost. AKI occurs in approximately 13.3 million people each year and is thought to contribute to about 1.7 million deaths each year [1]. One in five adults and one in three children worldwide have been reported to have AKI during a hospital admission [2]. While prior meta-analyses included data mainly from developing countries, a recent study using the Kidney Disease Improving Global Outcomes (KDIGO) definitions of AKI revealed an increasing number of studies from Africa, Latin America, and Southeast Asia. Additionally, the pooled incidence from lower-middle-income countries seems to be comparable to that of developed high-income countries. There are marked differences in access to and availability of renal replacement therapy when comparing high-income and low-income countries. Peritoneal dialysis (PD) is used more commonly than extracorporeal therapies when available in low-income countries [3]. Not surprisingly, a recent review investigating the outcomes of PD for AKI revealed that only 5 out of 24 included studies were from high-income countries, with most data having been published after the year 2000. Over the 1980s and 1990s, most developed countries had decreased research in acute PD coinciding with the increase in more technologically attractive options such as hemofiltration and hemodialysis [4].

Several advantages of acute PD result in it being used for AKI, specifically in places with limited resources (Table 1). Acute PD is technically simple, more cost effective, and requires less infrastructure than extracorporeal therapies. It avoids the need for vascular access and is especially advantageous in patients with bleeding diathesis. Additionally, it is a more physiologic, less inflammatory treatment than extracorporeal therapies, and its continuous nature results in gradual solute removal. Moreover, it results in hemodynamic tolerance with theoretical preservation of renal hemodynamics. On the other hand, several potential disadvantages have resulted in acute PD being utilized only in countries where resources do not allow the use of more advanced therapies. It is relatively contraindicated in patients with recent abdominal surgeries (with active peritoneum breach within 6 weeks), adynamic ileus, abdominal adhesions, and active peritonitis. Since volume and solute removal are slow and gradual, they may not be as efficient as other blood purification techniques for the treatment of life-threatening emergencies. Additionally, there is a risk of peritonitis complications as well as concerns for impaired respiratory mechanics from increased intra-abdominal pressure [5].

Several questions have been raised regarding utilization of acute PD, specifically related to outcomes and mortality differences compared to extracorporeal therapies, difference in renal recovery, ideal prescribed dialysis doses, infectious rates, and cost and economic implications. The aim of this review is to systematically address these concerns.

## 2. Mortality Outcomes

One of the earliest studies to investigate acute PD outcomes was a randomized control trial based in Vietnam which assessed the efficacy and safety of acute PD in patients with either severe falciparum malaria or sepsis. This study compared 34 patients undergoing continuous venovenous hemofiltration (CVVHF) with 36 patients undergoing manual PD. The study was terminated early due to high mortality (47%) in the PD group vs. 15% mortality in the CVVHF group (*p* < 0.005). Besides mortality, acute PD was also noted to be inferior to CVVHF in terms of the resolution of acidosis and renal failure. There were several concerns over the utilization of suboptimal PD techniques, namely, the usage of locally produced acetate-containing solutions, the insertion of rigid catheters associated with an increased risk of leaks, and the use of manual exchanges with an open drainage system. Notably, this study also had a high rate of culture-negative peritonitis [6]. Subsequent outcomes studies showed more promising outcomes in patients undergoing acute PD. An open prospective randomized study compared 25 patients undergoing manual continuous manual PD with 25 patients undergoing continuous venovenous hemodiafiltration (CVVHDF). Similar to the study mentioned prior, PD was conducted via a rigid catheter, and patients underwent 30 min PD fluid dwell times. The study investigated mortality rates as well as the composite correction of uremia, metabolic acidosis, fluid overload, and hyperkalemia. There was no significant difference in the mortality rate between the CVVHDF and PD groups (84% vs. 72%). Though the mortality rate was significantly high, it was comparable to rates in other centers in the region where patients with septic shock. Additionally, there was no difference noted in the composite correction of metabolic parameters; the PD group was noted to have improved correction of acidosis, whereas the CVVHDF group was noted to have faster correction of fluid overload [7]. Another randomized study conducted by a single center compared the outcomes of 63 patients undergoing continuous tidal PD with the outcomes of 62 patients undergoing CVVHDF. The prescribed doses were 25 L/day in the PD group and 30 mL/kg/h (delivered dose 23 mL/kg/h) in the CVVHDF arm. This study found the PD group to have improved 28-day survival compared to the CVVHDF group (69.8% vs. 46.8%) [8]. Two Brazilian studies have compared the outcomes of acute PD with those of hemodialysis in non-critically ill patients. The first study randomized 60 patients receiving high volume PD (HVPD) and compared them with 60 patients receiving daily hemodialysis (DHD). The prescribed Kt/V (dialysis adequacy) was 0.65 per day and 1.2 per session in the HVPD and DHD groups, respectively. The HVPD group received a daily prescription of 2 L exchanges with 35–50 min dwell time, resulting in 36–44 L of dialysis dose amounting to 18–22 exchanges per day. The DHD group received >3 h of dialysis 6 days a week. The study noted that despite having a higher Kt/V delivery in the DHD group, there was no difference in overall mortality rates between the two groups. Both groups also had similar outcomes in metabolic control, with results showing stabilization of metabolic parameters after the same number of dialysis sessions [9]. Comparing the HVPD group (61 patients) to the extended daily hemodialysis (EDD) group (82 patients), the HVPD group was prescribed treatment times and volumes that were similar to the prior study, whereas the EDD group was prescribed 6–8 h of HD, 6 days a week. Though the HD group was noted to have increased ultrafiltration and faster metabolic control, the two groups had similar overall mortality rates [10]. Though mortality outcomes were not significantly different, it must be noted that HVPD as a modality requires a large infrastructure including automated PD, electricity, personnel, and PD solutions, which may not be available in all situations.

## 3. Effects on Mechanical Ventilation

There are several concerns pertaining to increased intra-abdominal pressure with acute PD in critically ill patients undergoing mechanical ventilation for respiratory support. These include issues with impaired diaphragm mobilization, altered inspiratory and expiratory pressures, decreased pulmonary capacity, and worsening respiratory failure. Only one prospective cohort study has analyzed respiratory mechanics in patients undergoing mechanical ventilation and peritoneal dialysis for AKI. The study excluded patients with pre- or post-renal AKI, severe hemodynamic instability, tracheostomies, and patients requiring a positive end-expiratory pressure (PEEP) > 10 cm H_2_O. The patients underwent HVPD (36–44 L/day) and had established mechanical ventilation settings at a PEEP of 5 cm H_2_O and a tidal volume of 6 mL/kg. The parameters that were monitored were intra-abdominal pressure via intravesicular catheter, respiratory parameters (pulmonary compliance and respiratory system resistance), and oxygenation (PaO_2_/FiO_2_ and FiO_2_). The measurement of these parameters occurred during the first three days of dialysis treatment and included values taken during pre-dialysis, post-infusion and post-dialysis stages on day 1, as well as post-dialysis stages on days 2 and 3. The results demonstrated elevated intra-abdominal pressure at the initial evaluation (not uncommon during critical illness), which increased significantly after fluid instillation but resolved following drainage. Additionally, the increased intra-abdominal pressure remained at less than 15 mmHg (there are concerns for altered respiratory mechanics and airway pressures if pressure is >15 mm Hg) and did not contribute to worsening respiratory mechanics. There was a progressive increase in pulmonary compliance, possibly due to ultrafiltration and controlled intra-abdominal pressure. Furthermore, there was a significant increase in PaO_2_/FiO_2_ and FiO_2_ (improved respiratory parameters) [11].

## 4. Impact on Renal Recovery

Few studies have reported rates of and time to renal recovery, which has resulted in conflicting data. While one study reported an improved recovery of function as well as a shortened time for AKI resolution in the PD group (5 vs. 8 days) [8], another study reported overall similar rates of recovery; however, the PD arm was noted to have a shorter time to renal recovery (7 vs. 10 days) [9]. In contrast, recovery rates and time to AKI resolution were noted to be similar in the PD and HD groups in a third study that reported these outcomes [10].

## 5. Ideal Prescribed Dialysis Dose

Appropriate dosage for acute PD is poorly defined. Two studies have reported similar outcomes compared to hemodialysis with a Kt/V of 3.6 delivered weekly. They prescribed automated PD with a 35–50 min dwell time, undergoing 18–22 exchanges per day with 36–44 L of dialysis [9,10]. Only one randomized control trial has investigated the effect of PD dosage on outcomes for AKI. Patients were randomized to high intensity PD (Kt/V 0.8 per day) and low intensity PD (Kt/V 0.5 per day). The aim of the study was to compare mortality rates and metabolic control. There was a significant difference in the prescribed and delivered doses between the two groups; however, mortality rates, rates of renal recovery and time spent on dialysis were not different between the groups [12]. The current International Society of Peritoneal Dialysis (ISPD) guidelines recommend a weekly target Kt/V of 2.2. The prescription is recommended to be of 2 L dwell volume and 2 h dwell time, resulting in approximately 24 L of dialysis per day. The dwell cycles can be increased to 4–6 h following appropriate metabolic control [13].

## 6. Infectious and Mechanical Complications

Few studies have reported rates of infectious and mechanical complications in acute PD and data has been inconsistent. In a study utilizing an open drainage connect system as well as acetate-based solutions, high rates of culture-negative peritonitis were reported [6]. On the contrary, similar rates of peritonitis were observed in a study investigating high (12.9%) vs. low (13.3%) intensity PD [12]. Comparing PD with extracorporeal therapies, there were noted to be similar rates of peritonitis and blood stream infections in studies comparing acute PD (18%) to DHD (13%) as well as acute PD (16.3%) to EDD (19.5%) [9,10]. When comparing tidal PD to CVVHDF, decreased rates of catheter infections and catheter exchange were noted in the PD group [8].

## 7. Cost and Economic Implications

Not all studies have performed cost analyses that compare PD to extracorporeal therapies, and varying data is available. A study based out of India reported the cost of CCVHDF vs. PD disposables (including equipment) to be INR 7184 (USD 93) ± 1436 (USD 18) vs. INR 3009 (USD 39) ± 1643 (USD 21) (*p* < 0.01) [7]. This conflicts with data from an earlier study where the cost per PD survivor (USD 6950), using varying dextrose percentages injected into bags of Lactated Ringer’s solution, was significantly higher than the cost per CVVHF survivor (USD 2080) [6]. Cost implications are significant for middle- and lower-income countries where the cost of extracorporeal therapy equipment is expensive when compared to PD and lower costs are associated with local production of peritoneal fluid; costs are associated with imported, commercially available solutions.

## 8. Peritoneal Ultrafiltration for Refractory Heart Failure

Another reported use of PD in non-end stage kidney disease patients is peritoneal ultrafiltration for refractory heart failure. Diuretic refractory heart failure results in a significant burden on hospital admissions and has high 6-month (>50%) and 1-year (74%) mortality rates [14]. Briefly, the physiology of heart failure propagates a vicious cycle of cardio-renal deterioration with decreased renal perfusion, increased renin-angiotensin-aldosterone and sympathetic nervous system activation. These changes result in renal vasoconstriction, increased sodium and water reabsorption, decreased effect of atrial natriuretic peptide and increased effect of aldosterone, leading to diuretic resistance. Peritoneal ultrafiltration has been proposed as a treatment in home-based management of refractory heart failure and has been studied in small prospective and cohort studies.

Varying treatment regimens have been reported, which include manual daytime exchanges using a single icodextrin dwell, manual daytime exchanges using an overnight icodextrin dwell, and overnight automated exchanges for 2–4 nights each week [15]. Outcomes that have been studied include the New York Heart Association (NYHA) functional status, hospitalizations, quality of life, mortality, and echocardiographic parameters [15,16,17]. The largest retrospective cohort included 126 patients. The patients were reported to have had a reduction in body weight during the first 3 months of PD initiation, an improved left ventricular ejection fraction, a reduction in heart failure related hospitalizations, and a 1-year mortality of 42% [17]. Similar changes in hospitalizations have been noted in smaller cohort studies [15,16]. Regarding the NYHA functional class, it has been shown that at 1 year, 85% of patients had a reduction by at least 1 NYHA class, with echocardiographic parameters demonstrating an increased left ventricular ejection fraction and decreased pulmonary artery systolic pressures [15].

## 9. Acute PD Prescription

The latest ISPD guidelines for acute PD have detailed procedures and prescriptions that can be implemented in both resource-rich and resource-poor settings. The ideal recommended access is a flexible Tenkhoff catheter. Rigid catheters and drainage tubes are alternatives in resource-scarce areas. A closed system of fluid delivery and drainage is advised. The prescription can be either automated PD, which reduces nursing time but requires functional electricity, or manual PD, which is recommended in resource-poor settings. Standard dialysate is recommended when available; however, locally made solutions are an alternative where there are inadequate supplies. Though some studies have shown good outcomes with a targeted Kt/V of 3.5 (comparable to those of HD), other studies have shown similar outcomes with lower dialysis doses. Additionally, the volume of fluid required to achieve the higher clearances would be prohibitively expensive in certain countries. The acute PD prescription targets a weekly Kt/V of 2.1; its workflow is shown in Figure 1 [13].

## 10. Acute Peritoneal Dialysis during the COVID-19 Pandemic; Experiences and Lessons Learnt

In March 2020, with the onset of the COVID-19 pandemic in New York, acute PD became a necessity to deal with ongoing AKI cases. Mitigation strategies had not yet been developed and there were shortages of personal protective equipment. The incidence of AKI in hospitalized patients was higher than expected, and there was an increased demand for renal replacement therapies (15–20% of AKI patients). There was a scarcity of hemodialysis and continuous renal replacement therapy solutions, loss of dialysis personnel due to illness, and increased thrombosis of dialysis circuits. Logistical barriers included the provision of adequate dialysate for acute patients as well as established chronic PD patients. Additionally, access placement needed multidisciplinary planning given a shortage of operating room spaces (which had been converted into intensive care units), redeployment of personnel to critical care units, and uncertainty over laparoscopic surgery in patients with COVID-19 infections, as well as the feasibility of the transportation of critically ill patients. One center reported their experience in surgical open bedside catheter placement in ICUs. A series of 11 patients, all on mechanical ventilation, had catheters placed by a team of two surgeons and all had successful initiation of PD within 24 h [18].

A New York City PD consortium formed between four major academic centers reported their multicenter observational study of acute PD patients over a two-month period. This was the largest cohort describing acute PD during the early pandemic, with 94 patients, with PD being the initial renal replacement therapy modality for 56 patients. The majority of the patients (65%) were on manual PD exchanges. The main reasons for the transition from extracorporeal therapies (39%) were access thrombosis, supply shortages and nursing shortages. Of the patients, 76% were initiated on PD in ICU settings and 66% of PD catheters were placed by general surgeons. A total of 21% of the patients were switched to or supplemented with extracorporeal therapies due to catheter malfunction, supply shortages and metabolic derangements. This cohort of patients had 46% mortality and 22% renal recovery. Several pertinent positive outcomes were reported by this cohort. Of the prescribed PD volume, 86% was delivered, which is comparable to typical continuous renal replacement therapy. These patients had an improvement in serum potassium and bicarbonate levels. Though there was a high severity of illness in these patients, rates of death and renal recovery were comparable to AKI patients in other cohorts. Moreover, treatment goals were achieved despite 17% of patients being started on treatment in overflow makeshift intensive care units with curtailed access to water and electricity where extracorporeal therapies would not have been possible. Additionally, treatment goals were achieved despite a high proportion of obese patients [19].

A single center also published its experience of the utilization of acute PD while undergoing prone mechanical ventilation. Seven patients underwent bedside catheter placement followed by flushes with heparinized dialysate. Immediate initiation of manual exchanges with an incremental increase in dwell volume (maximum 2000 mL over 48 h) was followed to achieve a total daily volume of 10–16 L. During proning, the prescription was adjusted to a dwell volume of 1500 mL. A total of 71% of patients developed catheter leaks, which were resolved by decreasing the dwell volume to 500 mL for 12–24 h. Adequate ultrafiltration was achieved in all patients. An improvement in post-prone PaO_2_/FiO_2_ ratios was observed in all but one patient. Ventilator settings for all patients remained stable prior to and within 12–24 h of prone PD initiation. Given the need to minimize nursing staff exposure during the early pandemic, intra-abdominal pressures were not measured and would be interesting to pursue in future studies [20].

## 11. Future Directions

Very few randomized controlled trials (RCT) have compared acute PD with extracorporeal therapies. Currently, all of the analyzed RCTs have been conducted in India and Brazil, with sample sizes of less than 100 patients per group. There have been various methods and prescriptions of acute PD published to date, and there is a marked heterogeneity in the current RCTs. Several important outcomes measures need to be investigated and reported including the length of ICU and hospital stays. There is also a need for standardized reporting of technique, dosage, complications and cost.

## 12. Conclusions

Current data suggests that acute PD is an appropriate modality for renal replacement therapy, which has no significant difference in outcomes compared to extracorporeal therapies. Furthermore, acute PD has been recommended by the ISPD as a suitable method for renal replacement. The future of acute PD has been highlighted by the International Society of Nephrology’s 0 by 25 AKI initiative, which aims to have 0 preventable deaths from AKI by 2025. The Saving Young Lives Program established in 2012 has been instrumental in establishing acute PD programs in low-resource settings where dialysis was previously unavailable [21].

## Figures and Tables

**Figure 1 jcm-11-03270-f001:**
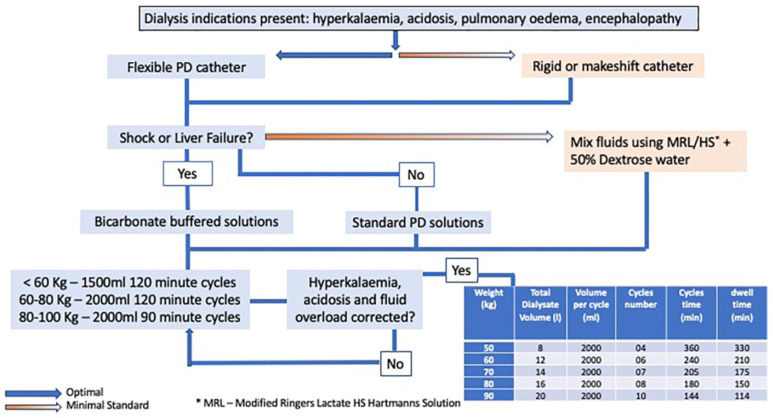
Acute peritoneal dialysis prescription [13].

**Table 1 jcm-11-03270-t001:** Advantages and disadvantages of Acute PD [5].

Advantages	Disadvantages
Technically simple	Contraindicated in recent abdominal surgery
Less infrastructure	Requires intact peritoneal cavity
Cost effective	May not be effective in severe acute pulmonary edema/hyperkalemia
Avoids vascular access	Peritonitis can occur
Biocompatible	Clearance and ultrafiltration unpredictable
Continuous renal replacement therapy	Concerns for hyperglycemia
Hemodynamic stability	Concerns for impaired respiratory mechanics
Gradual solute removal	Concerns for protein loss

## Data Availability

Not applicable.

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
