# Peer review of "Peritoneal Dialysis as a Renal Replacement Therapy Modality for Patients with Acute Kidney Injury"

_jcm, 2022, doi:10.3390/jcm11123270_

Round 1

Reviewer 1 Report

This paper reviewed the pros/cons (mostly pros) of PD as a renal replacement therapy for patients with AKI. This review describes it in detail and is easy to understand. However, there are several points which should be modified or emphasized, including the disadvantage of immediate PD for AKI such as the need/risk for PD catheter insertion and less effectiveness of PD for anuric patients.

Major concerns and questions for authors:

In the Introduction section

  1. Page 2, lines 45–48, and Table 1 – The author described that it is contraindicated to perform acute PD in patients with recent abdominal surgery. However, recent articles (Blood Purif 2021; 50: 328–35, Biomed Res Int. 2018; 2018: 1978765, and other statements) have suggested that PD is generally not contraindicated for patients with previous abdominal surgery. Therefore, the author should describe in detail whether PD is contraindicated only in acute setting, what type of surgeries are considered to be contraindication, and the period which we should consider “recent,” with some relevant references.
  2. Table 1 - I also consider that “intact peritoneal cavity” is not necessary in the acute phase.
  3. Table 1 – The disadvantage of acute PD includes the need to insert PD catheter (like insertion of vascular catheter or creation of AV fistula) and the risk of peri-catheter leak in the acute setting. Although the author mentioned these points in the “Acute Peritoneal Dialysis during Covid-19 …” section, these disadvantages should be more discussed in this Introduction section, or in other independent section (including preventive measures).

In the Mortality outcomes section

  1. Page 3, lines 96–97 – Delivered Kt/V is difficult to compare between HD and PD (because calculation method and homeostasis are totally different with each other).
  2. While HVPD is superior in terms of solute removal or prognosis, it is expected to increase the burden on patients and healthcare providers.

Additionally, considering the character of gradual removal of water and solute by PD and the importance of residual renal function on PD-related outcomes, only PD therapy may be ineffective in anuric patients especially with respect to the removal of middle molecules. The author should mention these points as disadvantages.

In the Effects on Mechanical Ventilation section

  1. Page 3, line 108 – HVPD (36–44 liters) … please clarify the liter per session, which is also important to consider the inverse effect on mechanical ventilation.

In the Ideal prescribed dialysis dose section

  1. Page 3, lines 133–134 – “Kt/V 0.8/session” “Kt/V 0.5/session” Do these “session” represent “day?”

In the Acute PD Prescription section

  1. Page 5, lines 194–195 – Why is the target Kt/V set as 2.1 by ISPD? This is far lower from 0.5–0.8 per day set in the clinical trials.

Author Response

I thank you for your thoughtful review, please note below 

1- The content has been edited to clarify abdominal surgery timeframe and type

2- This table has been adapted from a prior publication, however the text has been clarified. 

3- Noted, and more clarification provided in the text. 

4- Thank you for the suggestion, that has been deleted

5- Thank you, your point has been clarified

6- This has been clarified

7- This has also been clarified

8- The explanation has been elaborated

Reviewer 2 Report

The author presents a nice overview of peritoneal dialysis in patients with AKI and reviews relevant issues that arise in high income and low income settings with PD. 

  • line 30 - PD is more commonly used in high income or low income? unclear in the sentence as phrased.
  •  line 67 - missing the word "in" before patients and after outcomes
    Kt/V is not defined before use.
  • in section on effects of mechanical ventilation, more discussion on the the significance of 15 mmHg increase in intra-abdominal pressure and what values would be too much of an increase. Also more details about the increase in the PaO2/FiO2 and FiO2 would be helpful. 
  • impact on renal recovery section: discussion of actual time frames of recovery should be added
  • cost section: comparing two different types of currency is confusing, may be worth converting INR to $ for more direct comparison given $ for the second study. also the costs described, does that include the equipment necessary for each type of dialysis?
  • separate future directions section would be better than adding to conclusions.

Author Response

Thank you for your thoughtful critique. Pleas see below 

1- This point has been clarified 

2- thank you, this has been edited

3- this has been clarified as suggested

4- This has been edited 

5- edits made as suggested

6- this has been edited 

Round 2

Reviewer 1 Report

All of the issues that I pointed out were adequately revised. I have no additional comments.